# Early Guidance, Late Convergence: Hidden State Massive Values in Diffusion MLLMs

## Abstract

Diffusion multimodal large language models (dMLLMs) have emerged as a promising alternative to autoregressive generation, offering multi-token updates for finer control and faster inference. However, their internal mechanisms remain poorly understood, especially regarding how hidden states evolve across network layers and iterative denoising steps. In this work, we present the first systematic investigation of a striking phenomenon in dMLLMs: a small fraction of hidden state activations become extraordinarily large and consistently appear across layers and timesteps. We refer to this phenomenon as massive values. Our analysis reveals that in later layers and final diffusion steps, massive values align closely with the model's output semantics and confidence, directly influencing generation quality. In contrast, early layers and initial noisy steps produce massive values that are necessary to initiate generation and guide the global structure of content. Furthermore, using a sparse autoencoder to interpret hidden representations, we find that the evolution of these high-magnitude activations closely tracks the formation of output semantics. This indicates that the massive values are not just numerical outliers but are crucial drivers of the model's semantic generation process. Overall, our findings shed new light on the inner workings of dMLLMs and suggest potential strategies to improve their reliability and performance.

## 1 INTRODUCTION

Diffusion multimodal large language models (dMLLMs) have recently emerged as a new paradigm in the field of generative modeling. Compared to autoregressive models, dMLLMs perform multi-token updates within each denoising step, providing fine-grained control and more precise outputs(Li et al., 2025b; You et al., 2025; Yu et al., 2025b). Several studies have shown that dMLLMs achieve performance on many downstream tasks comparable to autoregressive models, while offering substantially faster inference(Israel et al., 2025; Wang et al., 2025; Yu et al., 2025a). Although dMLLMs have shown practical benefits, their internal mechanisms are still not well understood, in particular the evolution of hidden states during different layers and multi-step denoising. This knowledge gap limits our ability to further enhance both the reliability and the performance of these models(Jin et al., 2025; Sun et al., 2024; Gu et al., 2025).

Studies on LLMs have revealed a phenomenon that only a few activations attain significantly larger values than others and these activations are referred to as massive values (Sun et al., 2024; Jin et al., 2025). Recent work further shows that such massive values mainly arise in the query and key vectors of self-attention, but not in the value vectors (Jin et al., 2025). Several studies demonstrate that massive activations are central to the way attention is distributed in large models. In LLMs, they have been directly associated with attention sinks, with amplified token representations dominating the attention allocation(Gu et al., 2025; Yu et al., 2024; Sun et al., 2024). Similar effects have also been shown in multimodal models, where irrelevant vision tokens receive disproportionately high weights due to massive values(Kang et al., 2025). According to attention sinks caused by massive values, recent studies have explored different strategies to intervene on massive values and found that it can significantly improve multimodal model behavior(Kang et al., 2025; Zhu et al., 2025b). However, most existing studies focus on the influence of massive values at the attention layer rather than the hidden states. Besides, the internal mechanisms of dMLLMs have received little systematic investigation(Ben-Artzy & Schwartz, 2024; Jin et al., 2025). Motivated by this gap, we turn our attention to the hidden states of dMLLMs.

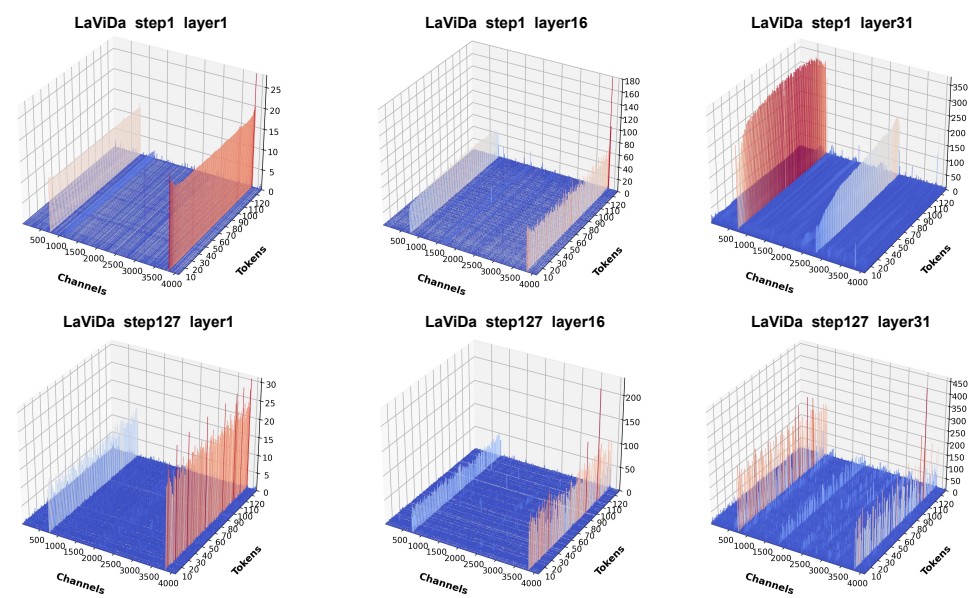

Figure 1: Visualization of massive hidden state values across layers and diffusion steps in LaVida. These massive values consistently emerge at different layers and timesteps, but their intensity and distribution vary with network depth and diffusion stage.

As is shown in Figure 1, we also observe the presence of **massive values** emerging consistently across layers and timesteps.To investigate the causes of this phenomenon and its role in generation, we conduct a systematic analysis from multiple perspectives. Specifically, our core observations can be summarized as follows: (i) **Layer-wise correlation.** We conduct correlation analysis and causal interventions on the hidden states across layers and reveal a strong correlation between massive values, token confidence, and the model outputs. Our analysis shows that massive values in deeper layers directly drive output semantics, whereas early-layer massive values are necessary for generation but mainly capture low-level features. (ii) **Step-wise dynamics.** By analyzing denoising stages, we observe distinct behavior between early steps and late steps, each exerting different influences on the model's output.Therefore, we find that massive values guide the global generation skeleton in early noisy steps, but later shift to regulating semantic convergence and directly aligning with output logits. (iii) **Semantic modeling.** Using a sparse autoencoder (SAE), we observe that massive values evolve in close synchrony with output semantics across the diffusion process. This provides evidence that massive values are not numerical artifacts but structural drivers of semantic formation.

In summary, the main contributions of this paper are:

- We present the first systematic study of massive values in the hidden states of dMLLMs, revealing their consistent presence across layers and timesteps and filling a key gap in understanding this new generative paradigm.

- We conduct a layer-wise analysis showing that massive values in deeper layers are strongly correlated with output confidence and directly determine generation quality, while early-layer peaks are necessary for generation but encode only low-level features.This finding reinforces our interpretation of layer functions.

- We perform a step-wise analysis across diffusion timesteps, demonstrating that massive values serve as global guiding signals in early noisy stages and later shift to regulating semantic convergence, closely aligning with final output logits.This finding highlights the distinctive influence of massive values in dMLLMs.

- We introduce a sparse autoencoder-based modeling approach that quantitatively validates the semantic contribution of massive values, offering new insight into the internal representations of dMLLMs.

## 2 RELATED WORK

### 2.1 MASSIVE VALUES IN LLMS

In recent years, a growing number of works have focused on masive values in LLMs, and have shown that they significantly affect model behavior and results. In quantization studies, several papers report that some dimensions in LLMs present massive values, which remain highly responsive at many sequence positions (Dettmers et al., 2022; Xiao et al., 2024; An et al., 2025; Son et al., 2024). Other work has shown that large activations can cause "attention sink" effects where certain tokens receive excessive focus (**?**). Meanwhile, in mechanism-level studies, researchers attempt to explain the causes of these massive values and propose methods to suppress or preserve them during quantization, such as OutlierTune's channel-wise quantization and SpinQuant's learned rotations (Wang et al., 2024; Liu et al., 2025).

Complementary interpretability research has also begun to examine which latent directions dominate LLM computation. Some work shows that individual hidden units do not represent single functions but encode multiple overlapping features (Gurnee et al., 2023; Elhage et al., 2022). To disentangle these, researchers use sparse autoencoders and linear probes to identify more "single-purpose" directions aligned with specific semantics. Others show that intervening can significantly alter generation, suggesting substantial control exerted by such hidden units (Zhang et al., 2025).

However, most prior work has focused on autoregressive LLMs and mainly examined massive values in attention layers, with limited exploration of hidden states. Motivated by this gap, we explore the role of massive values in the hidden states of dMLLMs.

### 2.2 DIFFUSION MULTIMODAL LARGE LANGUAGE MODELS

Recent work has moved diffusion from vision to language model.Surveys summarize this line and highlight parallel decoding and bidirectional context as key advantages over AR models (Yu et al., 2025b). On the language side, Diffusion-LM established a non-autoregressive denoising paradigm (Li et al., 2022). Large-scale variants such as LLaDA and its one-step distillation DLM-One further demonstrate competitive accuracy with improved sampling efficiency (Zhu et al., 2025a; Wu et al., 2025). On the multimodal side, LaViDA and LLaDA-V integrate visual encoders with diffusion backbones to achieve strong results on VQA and captioning benchmarks (Li et al., 2025b; You et al., 2025). Few studies have examined the internal structure and mechanisms of dMLLMs. Motivated by this gap, we explore the role of massive values in their hidden states to better explain the generation process and improve model reliability.

## 3 LAYER PERSPECTIVES ON THE IMPACT OF MASSIVE VALUES

In the above visualizations, we observe a common phenomenon: the emergence of extremely large activations (referred to as massive values) in the hidden layers. To understand their impact, we now focus on the fundamental unit of dMLLMs, the individual layer. By inspecting the hidden representations layer by layer, we aim to determine what roles these massive values play at different depths of the network.

### 3.1 DO MASSIVE VALUES IN HIDDEN LAYERS DRIVE THE MODEL'S OUTPUT?

We begin with a central question: **Do massive values in hidden layers actually influence the model's output?** To investigate this, we use the logit-lens and its tuned variant (Belrose et al., 2025), which map intermediate hidden states into vocabulary logits. This reveals layer-wise token predictions and allows us to directly test whether massive values align with and shape the model's eventual output.

Figure2 illustrates hidden-state behavior across layers and diffusion steps. Figure2(a) shows that early layers have relatively small peak activations, which grow steadily in deeper layers and reach their highest levels in the later layers. Figure2(b) exhibits a trend consistent with the overall pattern and Figure2(c) uses the logit lens to decode each layer's hidden state into a token prediction. In the shallow layers, the predicted token distributions are nearly uniform (high uncertainty). As the depth

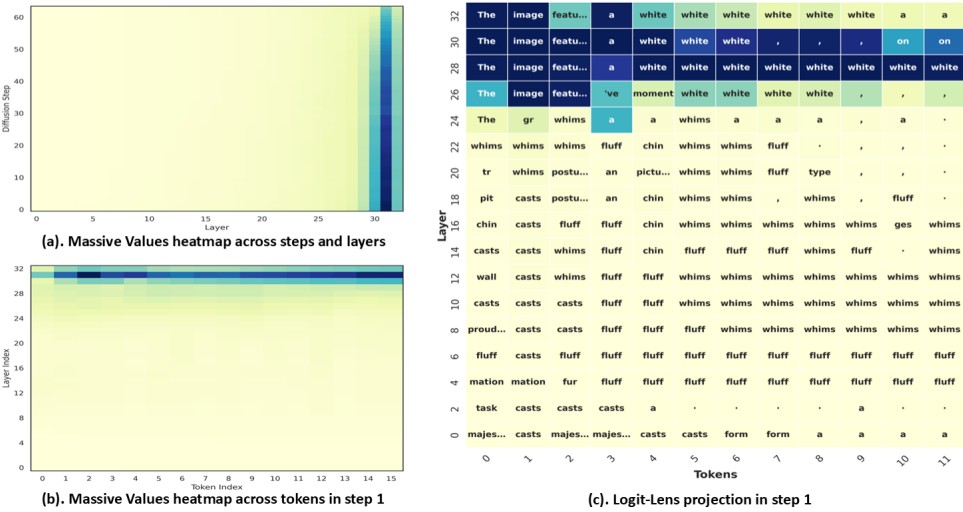

Figure 2: **Massive values visualizations in diffusion models.** (a) Step–layer activation map during denoising. (b) Token–layer activations for the first 16 tokens at step 1. (c) LogitLens projection of hidden states into vocabulary space at step 1, illustrating how semantic predictions emerge across layers.

increases, the model's intermediate predictions concentrate on semantically appropriate tokens, with much higher confidence in the correct outputs.

Comparing Figure2(a), Figure2(b) and Figure2(c) reveals a clear trend: as the average massive-value magnitude increases with layer depth, the model's confidence in its token predictions rises in tandem. This pattern is especially pronounced near layer 31, where the massive values peak and the model sharply boosts its confidence in the correct token. These observations show that in deeper layers, the emergence of massive values strongly correlates with the model's predictive confidence.

Thus, we can find that **massive values clearly correlate with token-level confidence.** This correlation raises a follow-up question: **do these massive values merely accompany confident predictions, or do they actively drive the model's output?**

## 3.2 DO MASSIVE VALUES PLAY A DECISIVE ROLE IN THE MODEL'S OUTPUT?

Although massive values clearly correlate with token-level confidence, we next ask whether they causally drive the model's output. To answer this, we introduce a metric called *Massive Values Channel Agreement*. This metric tests whether the channel with the massive values in a hidden state also corresponds to the channel that contributes the most to that state's output logit.

Formally, for diffusion step $s$, layer $\ell$, and token position $t$ with hidden state $h_{s,\ell,t} \in \mathbb{R}^D$, define the massive value channel index:

$$j^*(s,\ell,t) = \arg \max_{1 \le j \le D} \left| h_{s,\ell,t}[j] \right|$$

Next, we project $h_{s,\ell,t}$ into vocabulary logits using the logit lens, and define the locally predicted token:

$$\text{logits}_{s,\ell,t}(k) = h_{s,\ell,t} \cdot w_k \qquad \hat{y}(s,\ell,t) = \arg \max_k \text{logits}_{s,\ell,t}(k)$$

where $w_k$ is the embedding of token $k$, and $\hat{y}(s,\ell,t)$ is the token predicted by the layer-$\ell$ hidden state. For this predicted token $\hat{y}$, we compute each channel's contribution to the logit:

$$c_j(s,\ell,t) = h_{s,\ell,t}[j]\, w_{\hat{y}(s,\ell,t),\, j} \qquad j^{\text{contrib}}(s,\ell,t) = \arg \max_j c_j(s,\ell,t)$$

which is the product of channel $j$'s activation and the corresponding embedding weight for $\hat{y}$. Thus $j^{\text{contrib}}(s,\ell,t)$ is the index of the single channel that makes the largest logit contribution for the predicted token at that layer.

We define the Massive Values Channel Agreement at step $s$ and layer $\ell$ as the share of token positions whose massive-value channel coincides with the top logit contributor, and we average it over diffusion steps:

$$\text{Agree}(s,\ell) = \frac{1}{|\mathcal{T}_{s,\ell}|} \sum_{t \in \mathcal{T}_{s,\ell}} \mathbf{1}\big\{ j^*(s,\ell,t) = j^{\text{contrib}}(s,\ell,t) \big\}, \qquad \overline{\text{Agree}}(\ell) = \frac{1}{S} \sum_{s=0}^{S-1} \text{Agree}(s,\ell).$$

Here $\mathcal{T}_{s,\ell}$ denotes the set of valid token positions at step $s$. Under a null in which $j^*$ and $j^{\text{contrib}}$ are independent, the expected agreement is $1/D$. We report $\overline{\text{Agree}}(\ell)$ with 95% bootstrap confidence intervals across steps. Values substantially above $1/D$ indicate that massive-value channels systematically align with the readout-driving channels rather than arising by chance.

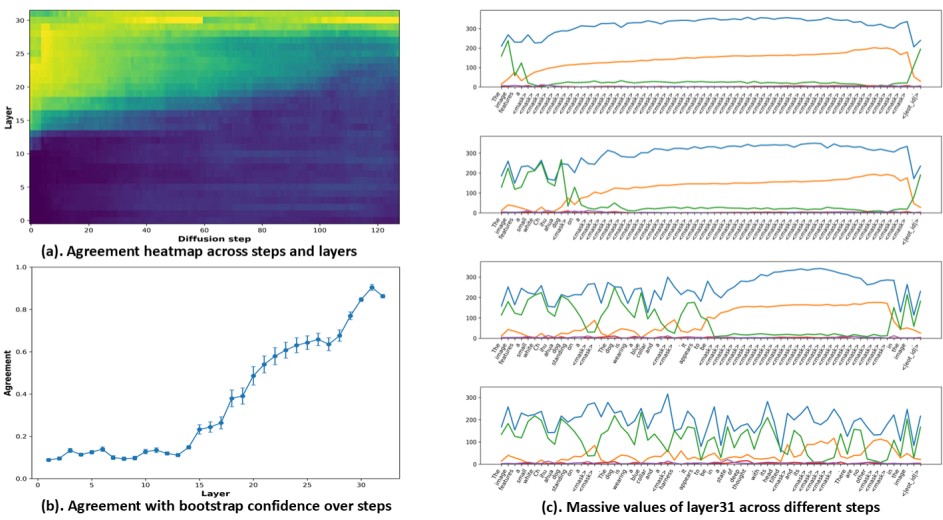

(a). Agreement heatmap across steps and layers

(b). Agreement with bootstrap confidence over steps

(c). Massive values of layer31 across different steps

Figure 3: **Agreement and massive-value dynamics across layers.** (a) Heatmap of channel–readout agreement $\text{Agree}(s,\ell)$ across diffusion steps and layers, showing a clear increase in deeper layers. (b) Layer-wise average agreement $\overline{\text{Agree}}(\ell)$ with 95% bootstrap confidence intervals, which rises sharply beyond mid-depth, far above the random baseline $1/D$. (c) Trajectories of massive values in layer 31 across steps. The green (channel 3848), blue (channel 753), and orange (channel 2823) curves correspond to dominant massive-value channels, while purple and red lines denote typical channels.

Figure 3(a) and Figure 3(b) confirms that massive values in deeper layers are not incidental spikes but systematically coincide with the channels that drive the readout. Agreement rises monotonically with depth, approaching near-perfect alignment in the final layers. This demonstrates that massive values evolve from low-level fluctuations in early layers to **decisive signals that mediate the transformation from internal representations to output predictions**.

These findings confirm that massive values have a direct influence on the model's output. We further hypothesize that **an activation needs to exceed a certain magnitude threshold to carry clear semantic significance.** Below that threshold, even a large activation might reflect only local fluctuations; beyond it, the activation acts as a gating signal that dramatically boosts the model's confidence in a particular token. Crossing this threshold marks a transition from a vague intermediate representation to a decisive, semantically clear output. Figure 3(c) highlights the dynamics of massive values at layer 31 across diffusion steps. However, a clear contrast emerges between masked tokens and generated tokens: while typical channels (purple, red) remain flat and uninformative, massive-value channels show strong and structured trajectories. In particular, channel 3848 (green) rises sharply around token generation, while channel 753 (blue) and channel 2823 (orange) remain small on generated tokens but dominate on non-token positions. **This asymmetry suggests that deep massive values act as selective gates for semantic outputs, consistent with a threshold-like mechanism. Importantly, different channels exhibit opposite behaviors: for some, semantic tokens emerge**

**only once activation surpasses a critical level, whereas for others, coherent outputs appear when activation remains below that level.**

Inspired by this observation, we conducted targeted interventions on the three dominant channels (Appendix A). When all three were zeroed, the model failed to produce any coherent output. More revealingly, ablating only channel 3848 was already sufficient to collapse generation completely, whereas removing either of the other two channels had little effect. **This indicates that channel 3848 is not just another large activation but plays a uniquely critical role, acting as a threshold-like gate that determines whether meaningful semantic generation can occur.**

To further validate this hypothesis, we intervened on massive-value channels by setting them to zero(Appendix B). When late-layer channels were ablated, the model's outputs collapsed into incoherent fragments, showing that these activations act as decisive switches in the final stage of generation. In contrast, zeroing early-layer channels also disrupted generation, but in a different manner: instead of directly preventing token selection, it removed the low-level features and global scaffold that deeper layers rely on to refine semantics. This distinction highlights a layered division of labor—early massive values provide essential structural grounding, while late massive values function as gating signals that commit internal representations into confident predictions—consistent with prior observations of shallow layers encoding local patterns and deeper layers capturing higher-level abstractions (Belinkov et al., 2019; Li et al., 2025a).

Overall, we argue that massive values in hidden layers are not mere numerical peaks: **they correlate with token confidence and are causally necessary for coherent generation.** Their roles can be summarized as follows:

- **Early layers:** Massive values are indispensable for capturing semantic cues and establishing the global representational scaffold.
- **Deep layers:** Massive values transition into decisional signals that directly govern token selection and reinforce semantic confidence.

## 4 STEP PERSPECTIVES ON THE IMPACT OF MASSIVE VALUES

In the previous section, we analyzed massive values from a layer-wise perspective at a fixed diffusion step. We now take a step-wise perspective, focusing on **how these extreme activations evolve as the diffusion process unfolds over time**.

### 4.1 HOW DO MASSIVE VALUES EVOLVE AS DIFFUSION NOISE DECAYS?

To investigate the behavior of massive values across diffusion timesteps, we tracked the distribution of token hidden-state magnitudes $|h_{s,l,t}|$ at each step $s$. For each step, we extracted various quantiles $Q_q(s)$ of this magnitude distribution and analyzed how they change as noise is gradually removed. We quantified each quantile's dependence on noise by fitting a linear trend $Q_q(s) \approx a(q) + b(q), \log \text{SNR}(s)$ and measuring the Spearman correlation $\rho(q)$ between $Q_q(s)$ and $\log \text{SNR}(s)$.

Figure 4 highlights the contrasting trends at two representative quantile levels. At a high-but-moderate quantile ($q = 0.9$), the activation magnitude clearly increases as $\log \text{SNR}$ increases (i.e. as diffusion noise decays), indicating that the bulk of activations grow stronger with less noise. Notably, at an extreme quantile ($q = 0.999$), this trend reverses: the top-tier activation magnitudes actually decrease as noise fades. In other words, the very largest outlier values are highest during the noisiest early steps and taper off later in diffusion. This quantile analysis reveals a clear two-phase dynamic in the diffusion process. In the early, a small number of channels dominate with extremely large activations that stand out above the noise. Later, those outlier activations subside while a broader set of channels attain moderate activation levels, collectively carrying the representation. In effect, the model's strategy shifts from an outlier-dominated regime in early diffusion to a bulk-dominated regime in the later steps. These observations naturally raise the question: **why would the model exhibit this two-phase behavior?**

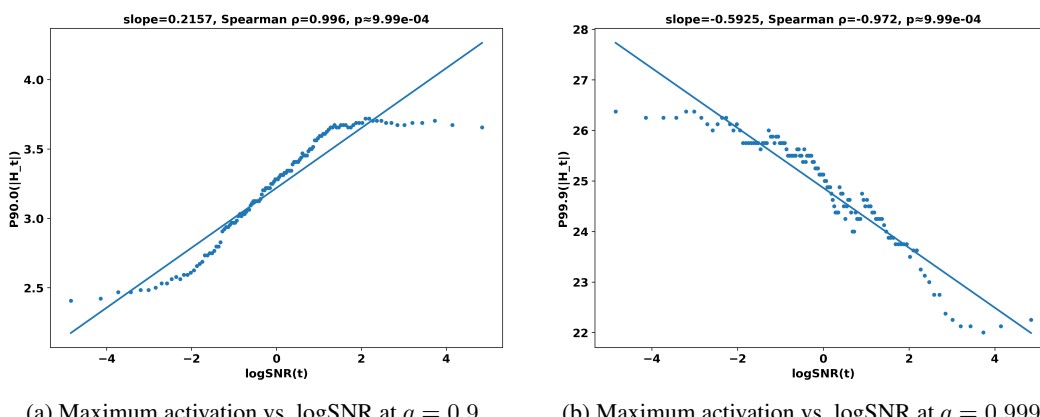

(a) Maximum activation vs. logSNR at $q = 0.9$.   (b) Maximum activation vs. logSNR at $q = 0.999$.

Figure 4: **Comparison of maximum activation–logSNR relationships at different quantiles.**

## 4.2 WHY WOULD THE MODEL OPERATE IN THIS TWO-PHASE MANNER?

The observed coarse-to-fine shift in massive values patterns can be understood by drawing a parallel to the denoising trajectory of diffusion models. During early timesteps, the model operates in a high-noise regime where only a few extreme activations can reliably rise above the stochastic background. These massive values thus act as global anchors, providing a directional signal that prevents the generation from collapsing into randomness. As noise diminishes, however, the model no longer needs a handful of extreme spikes to dominate; instead, a larger set of moderately strong activations collectively sustain and refine semantic content. This transition naturally mirrors the standard diffusion process, where generation first recovers coarse global structure and subsequently converges to fine-grained semantics as the signal-to-noise ratio improves (Nichol & Dhariwal, 2021; Karras et al., 2022; Song et al., 2021; Saharia et al., 2022). In this view, the two-phase behavior of massive values is not incidental but reflects stage-specific demands of the denoising dynamics: strong outliers are indispensable when noise is overwhelming, while more distributed activation patterns become sufficient for precise semantic alignment once the noise recedes.

To better understand the distinct roles of massive values across the diffusion trajectory, we performed controlled masking interventions on massive values at different diffusion stages. Specifically, we set the identified peak activations to zero under three conditions: early masking (first 32 steps), late masking (last 32 steps), and full masking (all steps), alongside a no-masking baseline (Appendix C).

As a result, when massive values are masked during the early diffusion steps, the model's generative process collapses, producing either no output or incoherent fragments devoid of semantic content, which indicates that these activations provide indispensable directional scaffolding for navigating the high-noise regime. By contrast, masking massive values only in later steps leads to milder degradation: the model continues to generate coherent and semantically meaningful sequences, though with reduced fluency, weaker confidence, and less fine-grained detail. These findings support our stage-specific hypothesis that **massive values play distinct roles across the diffusion process—acting as essential anchors that establish global structure in early steps and as stabilizing signals that refine and reinforce semantics in later steps.**

To further validate our findings and characterize step-wise update dynamics, we introduce the *residual energy* $E[\Delta](s)$, which quantifies the magnitude of representation updates across layers at step $s$. For hidden state $X_{s,l,t}^{(\cdot)} \in \mathbb{R}^D$ of token $t$ (with $(\cdot) \in \{\text{base}, \text{front}, \text{back}, \text{all}\}$ denoting baseline, early, late, or full ablation), we define

$$\Delta_{s,l,t}^{(\cdot)} = \|X_{s,l+1,t}^{(\cdot)} - X_{s,l,t}^{(\cdot)}\|_2, \qquad E[\Delta]^{(\cdot)}(s) = \mathbb{E}_{l,t}\big[\Delta_{s,l,t}^{(\cdot)}\big].$$

Here $\Delta_{s,l,t}^{(\cdot)}$ measures the update size for token $t$ from layer $l$ to $l+1$, and $E[\Delta]^{(\cdot)}(s)$ averages this across all layers and tokens to capture the overall update strength at step $s$.

Focusing on stage-specific effects, the residual-energy profiles corroborate our hypothesis. **Early phase:** the large spikes under early ablation indicate that, without massive values, the model

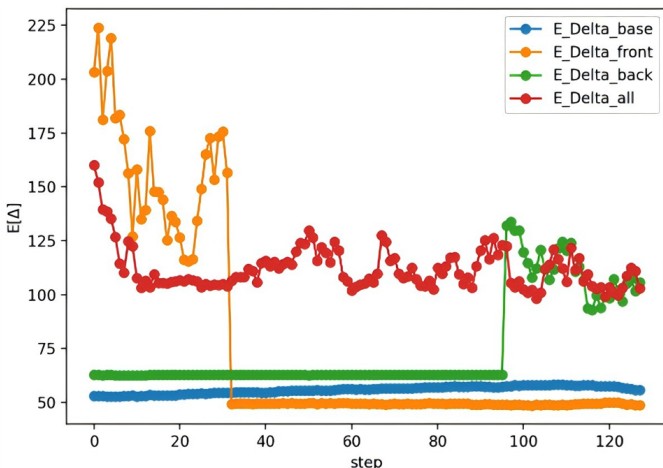

Figure 5: **Residual energy $E[\Delta]$ across diffusion steps under different ablation conditions.** Early ablation causes sharp initial spikes, late ablation induces abrupt rises near the end, full ablation remains elevated throughout, while the baseline stays stable and low.

lacks a stable directional scaffold and must execute oversized, volatile updates to orient the trajectory—consistent with massive values acting as early guiding signals. **Late phase:** the abrupt rise under late ablation shows that, once coarse structure is in place, massive values remain essential to semantic convergence; removing them forces costly corrective updates to maintain token-level consistency.

In summary, the evidence shows that massive values across diffusion timesteps are not random artifacts: **they serve as stage-specific control signals, first anchoring global structure and later reinforcing token-level confidence and coherence.** Their functional roles can be distilled as:

- **Early steps:** Massive values act as directional beacons, guiding the model to establish a global skeleton under heavy noise and preventing collapse into incoherence.
- **Late steps:** Massive values primarily support semantic convergence, consolidating token confidence and enhancing the fluency and precision of outputs.

## 5 GLOBAL PERSPECTIVES ON THE IMPACT OF MASSIVE VALUES

Having established that massive values directly influence model outputs, we now examine them from a *global perspective*. In particular, we ask whether these directions, which appear locally as single-channel spikes, also emerge as stable and reusable semantic axes when analyzed with sparse autoencoders (SAEs) (O'Neill et al., 2024; Deng et al., 2025; Gurnee et al., 2023). In other words, **do the channels that spike to massive values represent meaningful latent features?** This question is inspired by the observation that such dominant activations resemble the sparse, axis-aligned features learned by SAEs. To probe this, we analyze the model's hidden representations from an SAE perspective and test whether the learned dictionary atoms align with the model's massive-value directions.

In our experiments, we collect hidden vectors ($D = 4096$) from all layers and time steps, standardize them to remove scale differences, and extract an extreme subset where a single channel dominates. Intuitively, these states correspond to one massive channel carrying most of the representation's energy (Details in Appendix D.1). We then train a one-layer sparse autoencoder with ReLU activations and sparsity regularization. At inference, we reconstruct samples with a limited number of active atoms ($k \in \{1, 5, 10\}$) to test whether the SAE captures the massive-value phenomenon. (Details in Appendix D.2)

To assess alignment between SAE atoms and massive-value directions, we use four measures: (1) *Top-$k$ hit rate*, the fraction of samples where the SAE's top-$k$ active atoms include the max channel

of the original representation; (2) *Purity-L1*, the degree to which an atom's weight mass is concentrated on a single input dimension (higher = more axis-aligned); (3) *Active-#*, the average number of atoms used per reconstruction (lower = more compact representations); (4) *MSE*, the reconstruction error in $z$-score space (lower = more accurate reconstructions).

| Group | Top-1 | Top-5 | Top-10 | Purity-L1 | Active-# | MSE |
|---|---|---|---|---|---|---|
| global_on_test | 0.00036 | 0.02876 | 0.03671 | $4.78 \times 10^{-4}$ | 2680.24 | 89.14 |
| global_on_ext_test | 0.00031 | 0.10085 | 0.10085 | $1.23 \times 10^{-3}$ | 41.34 | $6.56 \times 10^{-3}$ |

Table 1: **SAE performance on the regular test set vs. the extreme subset.**

Table 1 shows that on the full test set, SAE features are broad and redundant: Top-5/10 hit rates are only 2.9%/3.7%, Purity-L1 is nearly zero, each sample activates $\sim 2680$ atoms, and reconstruction error is high (MSE $\approx 89$). This indicates highly entangled representations with no single dominant direction, making it difficult for the autoencoder to isolate interpretable structure. By contrast, the extreme subset shows sharp improvements: Top-5/10 hit rates rise to 10.1%, Purity-L1 triples, Active-# drops to $\sim 41$, and MSE falls by four orders of magnitude ($6.56 \times 10^{-3}$). Beyond the raw numbers, these differences highlight a fundamental representational shift. In regular states, information is dispersed across many overlapping channels, forcing the SAE to use thousands of atoms to reconstruct each vector with poor fidelity. In extreme states, however, energy is concentrated along a single channel, enabling the SAE to recover the representation with just a few atoms and minimal error. This pattern suggests that massive values carve out axis-aligned semantic directions within the latent space: when they dominate, the representation becomes sparse, interpretable, and efficient. In other words, **massive values provide the model with reusable semantic axes that act as compact building blocks for generation, rather than being incidental numerical outliers.**

Based on the above experiments, we conclude that massive values represent meaningful latent features and play an essential semantic role in the model's representation space:

- **Global signals:** Massive values manifest as stable semantic directions, aligning with sparse autoencoder features and serving as reusable latent axes in the representation space.

## 6 CONCLUSION

In this work, we provided the first systematic study of hidden-states massive values in diffusion-based multimodal language models. Our investigation proceeded from the smallest unit of analysis to the most global view. At the layer level, we showed that massive values correlate with token confidence: shallow peaks scaffold early representations, while deep-layer spikes become decisive signals that directly govern output predictions. At the step level, we uncovered a coarse-to-fine trajectory across the diffusion process: early massive values act as global beacons that orient generation under heavy noise, whereas late-stage activations stabilize semantic convergence and refine confidence. From a global representational perspective, sparse autoencoder analysis revealed that massive values align with stable, axis-aligned semantic directions reused across examples, forming a backbone vocabulary of latent features. Together, these perspectives converge to a unified insight: massive values function as structural signals that orchestrate the formation of global structure, refine intermediate representations, and ground semantics throughout the generation process in dMLLMs.

## 7 LIMITATION

One limitation of our study is that we restrict the analysis to hidden-state dynamics without explicitly examining how massive values interact with attention mechanisms. Since prior work has connected large activations to attention sinks, integrating hidden-state and attention-path analyses in future work could provide a more complete picture of how extreme activations shape both representations and attention allocation.

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

## USE OF LLM

The authors used generative AI tools (e.g., Grammarly, ChatGPT) solely for grammar checking and language polishing of the manuscript. All technical content, experimental design, data analysis, and conclusions were generated and verified exclusively by the human authors. The use of AI tools does not affect the originality or authorship of this work.

## A    MASK CASE STUDY IN CHANNEL-WISE

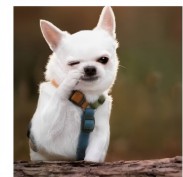

You are a helpful language and vision assistant. You are able to understand the visual content that the user provides, and assist the user with a variety of tasks using natural language. Please describe the image in detail.

**Normal output:**
The image features a white Chihuahua dog, standing on a wooden deck. The dog is wearing a blue collar and a green bandana around its neck, adding a touch of color to its white fur. The dog's head is tilted slightly, and it appears to be looking at something out of the frame, creating a sense of curiosity or interest. The background is blurred, but it seems to be an outdoor setting, possibly a park or a wooded area, with a hint of trees and foliage. The overall composition of the image, with the dog in the foreground and the blurred background, creates a sense of depth and perspective.

**Mask on Channel 3848:**
))))),)))))))).,. A,, 0AP

**Mask on Channel 753 :**
The image features a white Chihuahua dog, standing on a wooden deck. The Chihuahua is wearing a blue collar, a striking contrast to its white fur. The dog's head is tilted slightly, and it appears to be looking at something out of the frame, creating a sense of curiosity or interest. The trunk of a tree can be seen in the background, adding a touch of nature to the scene. The blurred background draws the viewer's attention to the Chihuahua, making it the focal point of the image.

**Mask on Channel 2823:**
The image portrays a charming white Chihuahua puppy, perched atop a sturdy tree stump. The puppy gazes directly into the camera, its head tilted downwards as if it\'s pondering contemplating contemplating contemplating contemplating stump stump stump. A blue collar adorned with a tag that reads "whim whim whim whimsy whim whim whim whim whim whim reads "puppy puppy puppy puppy collar hangs loosely puppy puppy collar, mirroring its perch atop the tree stump. A well-maintained grassy lawn surrounds the tree stump, providing a serene backdrop for the adorable puppy. The entire scene exudes tranquility and a harmonious coexistence with our canine companions.

Figure 6: **Channel-wise mask analysis of LaViDa, Case 1.**

In Figures 6 and 7, we conduct channel-wise masking interventions to examine whether massive-value channels causally affect semantic generation. Starting from the normal output, we ablate a single dominant channel (e.g., 3848, 753, 2823) while keeping all others unchanged. The results exhibit clear divergences: some channels yield degenerate strings or nonsensical fragments, others retain partial grammaticality but lose semantic fidelity, and certain channels disproportionately amplify irrelevant patterns. These findings highlight two key insights. First, late-layer massive-value channels act as decisive gates: their removal collapses output semantics, confirming a causal role beyond simple correlation. Second, different massive-value directions correspond to distinct semantic functions, such as object grounding versus structural consistency, reinforcing our claim that these channels represent reusable, interpretable latent features rather than incidental numerical outliers.

## B    MASK CASE STUDY IN LAYER-WISE

702
703
704
705
706
707
708
709
710
711
712
713
714
715
716
717
718
719
720
721
722
723

You are a helpful language and vision assistant. You are able to understand the visual content that the user provides, and assist the user with a variety of tasks using natural language. Please describe the image in detail.

**Normal output:**
In the image, a man and a woman are standing on a bridge, their faces illuminated by the glow of fireworks. The man, clad in a black shirt, is looking down at the woman, who is dressed in a pink dress. The fireworks, a mix of red and white, are exploding in the sky, creating a vibrant backdrop to the scene. The cityscape in the distance is a blend of gray and blue, with buildings and billboards dotting the horizon, reflecting the bright light of the fireworks. The image exudes a sense of joy and excitement, capturing a moment of celebration in the heart of the city.

**Mask on Channel 3848:**
......))))))......,,,a......

**Mask on Channel 753 :**
In the image, a man and a woman are standing on a bridge, their faces illuminated by the glow of fireworks. The man, clad in a black shirt, is looking down at the woman, who is dressed in a a a. The sky lights up with red and white fireworks, forming a brilliant background to the moment. The fireworks illuminate the gray-blue skyline dotted with buildings and signs. The image exudes a sense of joy and excitement, capturing a moment of celebration in the heart of the city.

**Mask on Channel 2823:**
In the image, a man and a woman are standing standing standing standing standing standing standing standing, their faces illuminated by the glow of fireworks. The man, clad in a black shirt, is looking down at the woman, who is dressed in in in in. The fireworks, a mix of red and white, are exploding in the sky, creating a vibrant vibrant vibrant vibrant vibrant. The cityscape in the distance is a blend of gray and blue, with buildings and billboards dotting the horizon,,,,,,,. The image shows a a sense of excitement,,,,,,, in the heart of the city.

Figure 7: **Channel-wise mask analysis of LaViDa, Case 2.**

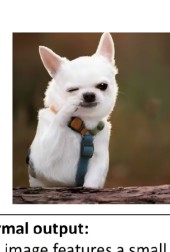

You are a helpful language and vision assistant. You are able to understand the visual content that the user provides, and assist the user with a variety of tasks using natural language. Please describe the image in detail.

**Normal output:**
The image features a small white Chihuahua dog, standing on a wooden bench. The dog is wearing a blue collar and a green bandana around its neck. It is looking directly at the camera, giving a sense of connection with the viewer. The dog's head is slightly tilted, and it appears to be in the middle of a thought, as if it's trying to understand the camera. The background is blurred, but it seems to be an outdoor setting, possibly a park or a garden. The dog's position on the bench, along with its direct gaze towards the camera, makes it a focal point in the image.

**Mask First 1 layer each step Output:**
.....................a................c.................................\................................

**Mask First 4 layer each step Output:**
.....................a................c.................................\................................

**Mask First 8 layer each step Output:**
...........................................\..............................................................

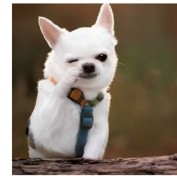

You are a helpful language and vision assistant. You are able to understand the visual content that the user provides, and assist the user with a variety of tasks using natural language. Please describe the image in detail.

**Normal output:**
The image features a small white Chihuahua dog, standing on a wooden bench. The dog is wearing a blue collar and a green bandana around its neck. It is looking directly at the camera, giving a sense of connection with the viewer. The dog's head is slightly tilted, and it appears to be in the middle of a thought, as if it's trying to understand the camera. The background is blurred, but it seems to be an outdoor setting, possibly a park or a garden. The dog's position on the bench, along with its direct gaze towards the camera, makes it a focal point in the image.

**Mask Back 1 layer each step Output:**
................................................................................................................

**Mask Back 4 layer each step Output:**
................................................................................................................

**Mask Back 8 layer each step Output:**
................................................................................................................

Figure 8: **Layer-wise mask analysis of LaViDa Case 1.**

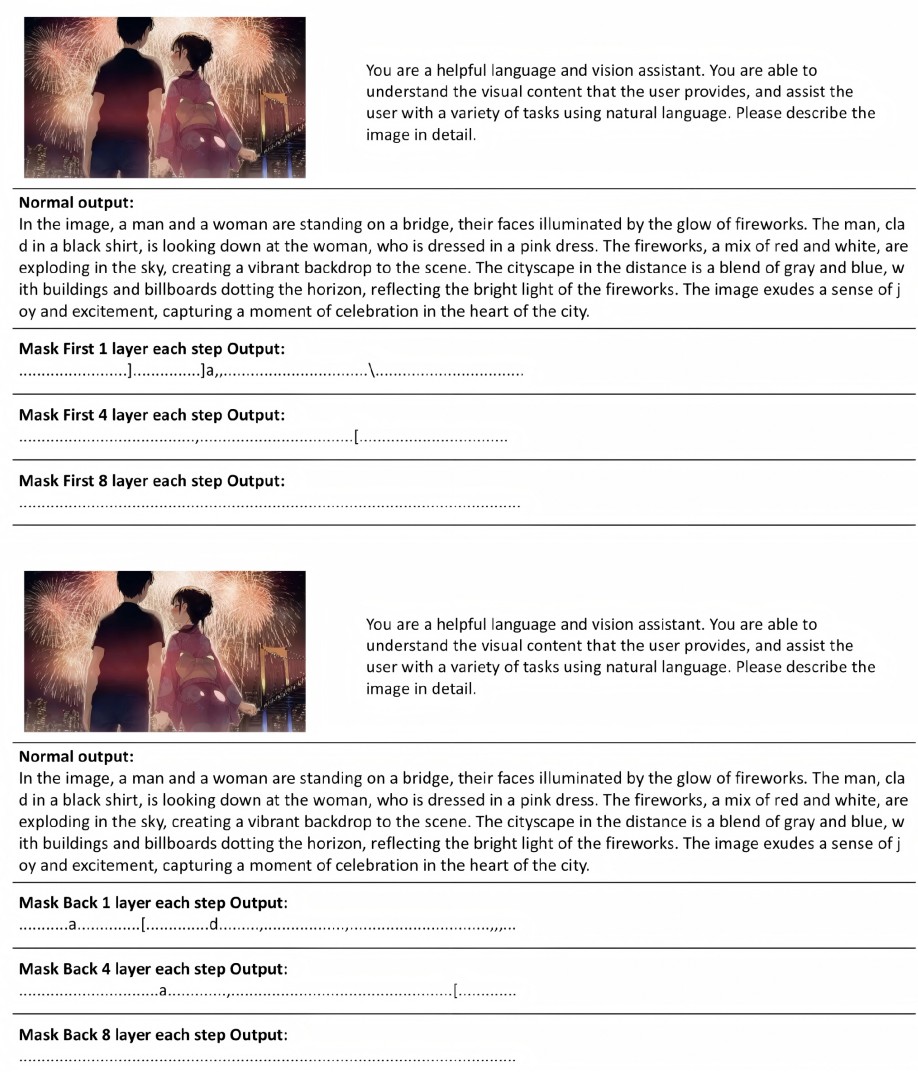

Figure 9: **Layer-wise mask analysis of LaViDa Case 2.**

We intervene on different depths of the model by masking massive values of the first or last $k$ layers at each diffusion step and examine the resulting outputs as Figure8 and Figure9. While the normal model produces coherent and detailed captions of the input Chihuahua image, progressively masking more early layers or late layers leads to degraded outputs. Early-layer masking primarily disrupts low-level grounding and progressively erodes semantic fidelity, whereas late-layer masking suppresses high-level alignment and yields near-empty responses. This contrast highlights that early layers provide essential scaffolding for token formation, while later layers consolidate semantics and preserve output coherence.

## C    MASK CASE STUDY IN STEP-WISE

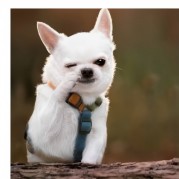

You are a helpful language and vision assistant. You are able to understand the visual content that the user provides, and assist the user with a variety of tasks using natural language. Please describe the image in detail.

**Normal output:**
The image features a small white Chihuahua dog, standing on a wooden bench. The dog is wearing a blue collar and a green bandana around its neck. It is looking directly at the camera, giving a sense of connection with the viewer. The dog's head is slightly tilted, and it appears to be in the middle of a thought, as if it's trying to understand the camera. The background is blurred, but it seems to be an outdoor setting, possibly a park or a garden. The dog's position on the bench, along with its direct gaze towards the camera, makes it a focal point in the image.

**Mask First 2 Step Output:**
<start of description>\nThe, features a, white

**Mask First 8 Step Output:**
........................a....................................................g.................................

**Mask First 16 Step Output:**
.......................................................................................................................

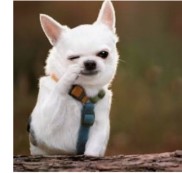

You are a helpful language and vision assistant. You are able to understand the visual content that the user provides, and assist the user with a variety of tasks using natural language. Please describe the image in detail.

**Normal output:**
The image features a small white Chihuahua dog, standing on a wooden bench. The dog is wearing a blue collar and a green bandana around its neck. It is looking directly at the camera, giving a sense of connection with the viewer. The dog's head is slightly tilted, and it appears to be in the middle of a thought, as if it's trying to understand the camera. The background is blurred, but it seems to be an outdoor setting, possibly a park or a garden. The dog's position on the bench, along with its direct gaze towards the camera, makes it a focal point in the image.

**Mask Back 2 Step Output:**
The image features a white Chihuahua dog, standing on a wooden bench. The dog is wearing a blue collar and a harness, suggesting it might be a pet or a show dog. The dog's head is tilted slightly, and it appears to be looking at something out of the frame. The background is blurred, but it seems to be an outdoor setting, possibly a park or a garden. The dog's position on the bench and its gaze suggest that it might be waiting for someone or observing something in the distance. The image does not contain any text. The overall composition of the image suggests a calm and peaceful atmosphere.

**Mask Back 8 Step Output:**
The image features a white Chihuahua dog, standing on a wooden. The dog is wearing a blue collar and a green bandana around its neck. It is looking directly at the camera, giving a sense of connection with the viewer. The dog's head is slightly tilted, and it appears to be in a state of deep thought or contemplation. The background of the image is blurred, suggesting that the dog is the main subject of the photo. The image does not contain any text. The relative position of the dog to the camera and the focus) that the dog is the) in the image.

**Mask Back 16 Step Output:**
The image features a white Chihuahua dog, standing on a wooden. The dog is wearing a blue collar and a green bandana around its neck. It is looking directly at the camera, giving a sense of connection with the viewer. The dog's head is slightly tilted, and it appears to be in a state of deep thought or contemplation. The background of the image is blurred, suggesting that the dog is the main subject of the photo. The image does not contain any text. The relative position of the dog to the camera,\n,, C,, in the image.

Figure 10: **Step-wise ablation analysis of LaViDa Case 1.**

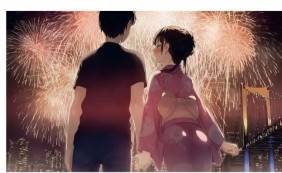

You are a helpful language and vision assistant. You are able to understand the visual content that the user provides, and assist the user with a variety of tasks using natural language. Please describe the image in detail.

**Normal output:**
In the image, a man and a woman are standing on a bridge, their faces illuminated by the glow of fireworks. The man, clad in a black shirt, is looking down at the woman, who is dressed in a pink dress. The fireworks, a mix of red and white, are exploding in the sky, creating a vibrant backdrop to the scene. The cityscape in the distance is a blend of gray and blue, with buildings and billboards dotting the horizon, reflecting the bright light of the fireworks. The image exudes a sense of joy and excitement, capturing a moment of celebration in the heart of the city.

**Mask First 2 Step Output:**
<start of description>\n <start of description>\n....

**Mask First 8 Step Output:**
...........................................................................................................................

**Mask First 16 Step Output:**
...........................................................................................................................

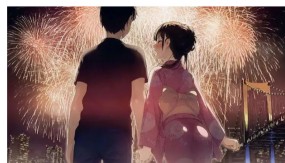

You are a helpful language and vision assistant. You are able to understand the visual content that the user provides, and assist the user with a variety of tasks using natural language. Please describe the image in detail.

**Normal output:**
In the image, a man and a woman are standing on a bridge, their faces illuminated by the glow of fireworks. The man, clad in a black shirt, is looking down at the woman, who is dressed in a pink dress. The fireworks, a mix of red and white, are exploding in the sky, creating a vibrant backdrop to the scene. The cityscape in the distance is a blend of gray and blue, with buildings and billboards dotting the horizon, reflecting the bright light of the fireworks. The image exudes a sense of joy and excitement, capturing a moment of celebration in the heart of the city.

**Mask Back 2 Step Output:**
In the image, a man and a woman are standing on a bridge, their faces illuminated by the glow of fireworks. The man, clad in a black shirt, is looking down at the woman, who is dressed in a pink dress. They are both facing towards the camera, their attention drawn to the vibrant display of fireworks in the background. The fireworks, a mix of red and white hues, add a festive touch to the scene. The bridge they stand on is visible in the distance, adding a sense of depth to the image. The image captures a moment of joy and C, set against the backdrop of a city celebration.

**Mask Back 8 Step Output:**
In the image, a man and a woman are standing on a bridge, their faces illuminated by the sparkle of fireworks. The man, clad in a black shirt, is looking down at the woman, who is dressed in a pink dress. The fireworks, a mix of red and white, are exploding in the sky, creating a vibrant backdrop to the scene. The cityscape in the distance is a blend of, and,, with buildings an d,,. the,, reflecting the bright light of the fireworks. The image exudes a sense of joy and, capturing a moment of celebration in the heart of the city.

**Mask Back 16 Step Output:**
In the image, a man and a woman are standing on a bridge, their faces illuminated by the glow of fireworks. The man, clad in a black shirt, is looking down at the woman, who is dressed in a pink dress. They are both facing towards the camera, their attention drawn to the vibrant display of fireworks in the background. The fireworks, a mix of red and white hues, add a festive touch to the scene. The bridge under them is a gray, providing a stark contrast to the colorful spectacle above. The image captures a,,,,,,,,,.

Figure 11: **Step-wise ablation analysis of LaViDa Case 2.**

We intervene on different diffusion phases by masking massive values of the first or last $k$ steps and compare the generated outputs as Figure10 and Figure11. Masking early steps disrupts the initial scaffolding of token formation, yielding fragmented or incoherent strings when more steps are suppressed. Masking later steps preserves basic structure but gradually degrades semantic coherence and completeness, leading to truncated or repetitive captions. This contrast highlights that early steps establish coarse semantic grounding, while later steps refine and consolidate the final description.

# D  SPARSE AUTOENCODER (SAE) ANALYSIS AND TRAINING DETAILS

## D.1  FORMULATION AND INFERENCE

In this subsection, we provide details of the sparse autoencoder (SAE) used in our analysis.

For each diffusion step $s \in \{0, \dots, S-1\}$, layer $\ell \in \{1, \dots, L\}$, and token $t$, we collect the hidden vector $h_{s,\ell,t} \in \mathbb{R}^D$ ($D = 4096$) and standardize it per-dimension using statistics from the training pool:

$$z_{s,\ell,t}[j] \;=\; \frac{h_{s,\ell,t}[j] - \mu_j}{\sigma_j + \varepsilon}, \qquad j = 1, \dots, D,$$

with $\varepsilon$ for numerical stability. We denote a standardized sample by $z \in \mathbb{R}^D$. We use a one-layer SAE with ReLU codes and an overcomplete dictionary of $M$ atoms ($M \geq D$):

$$a \;=\; \mathrm{ReLU}(W_e z + b_e) \in \mathbb{R}^M_{\geq 0}, \qquad \hat{z} \;=\; W_d a + b_d \in \mathbb{R}^D,$$

where $W_e \in \mathbb{R}^{M \times D}$, $b_e \in \mathbb{R}^M$, $W_d \in \mathbb{R}^{D \times M}$, $b_d \in \mathbb{R}^D$. We do not tie weights ($W_d \neq W_e^\top$). Given a minibatch $\{z^{(n)}\}_{n=1}^B$, the loss is

$$\mathcal{L} \;=\; \frac{1}{B}\sum_{n=1}^B \big\| z^{(n)} - \hat{z}^{(n)} \big\|_2^2 \;+\; \lambda \frac{1}{B}\sum_{n=1}^B \big\| a^{(n)} \big\|_1 \;+\; \alpha\big(\|W_e\|_F^2 + \|W_d\|_F^2\big).$$

After each update, we optionally project decoder atoms $d_i = W_d[:,i]$ to the unit $\ell_2$ ball: $d_i \leftarrow d_i / \max(1, \|d_i\|_2)$.

For a sample $z$, let $j^\star = \arg\max_j |z[j]|$, the dominance ratio $r(z) = |z[j^\star]|/\|z\|_2$, and the gap $\rho(z) = |z[j^\star]|/(\max_{j \neq j^\star} |z[j]| + \epsilon)$. We select an extreme subset by

$$r(z) \geq \tau_{\mathrm{dom}}, \quad \rho(z) \geq \tau_{\mathrm{gap}}, \quad |z[j^\star]| \geq q_{\mathrm{ext}},$$

where $q_{\mathrm{ext}}$ is a high global quantile (e.g., 99.5–99.9th); results are robust for $\tau_{\mathrm{dom}} \in [0.35, 0.50]$, $\tau_{\mathrm{gap}} \in [1.5, 2.0]$.

At test time, we use unconstrained codes $a$ for reconstruction and sparsity reporting, and form *top-k* codes by keeping the $k$ largest entries of $a$ ($k \in \{1, 5, 10\}$), yielding $\hat{z}_k = W_d a_{(k)} + b_d$. Let $\mathrm{dom}(i) = \arg\max_j |W_d[j,i]|$ be the dominant input channel of atom $i$. For a sample with maximal channel $j^\star$, the *Top-k hit* is 1 if $\exists i$ among the top-$k$ active atoms with $\mathrm{dom}(i) = j^\star$. We also report:

$$\mathrm{Purity\text{-}L1}(i) = \frac{\max_j |W_d[j,i]|}{\sum_j |W_d[j,i]|}, \quad \mathrm{Active\text{-}\#} = \|a\|_0, \quad \mathrm{MSE} = \|z - \hat{z}\|_2^2.$$

## D.2  TRAINING PROTOCOL AND HYPERPARAMETERS

We first aggregate standardized hidden states from all layers and diffusion steps, uniformly subsampling $(s, \ell, t)$ to avoid imbalance. Based on these, we then build train/validation/test splits and additionally evaluate on the extreme subset defined above. For the autoencoder, we adopt overcomplete dictionaries with $M \in \{2D, 4D, 8D\}$, and we observe that qualitative trends remain consistent across $M$. Next, we train the model with Adam (batch size $B$ chosen according to hardware, learning rate on the order of $10^{-3}$, weight decay $\alpha$ in ($[10^{-6}, 10^{-5}]$), using global-norm gradient clipping at 1.0 and optionally applying cosine decay. Since inputs are standardized, no batch normalization is used. During training, we further sweep $\lambda$ on a log grid (e.g., $10^{-4}$–$10^{-2}$) and select the configuration achieving the smallest validation MSE subject to target sparsity on the extreme subset and stable reconstruction. To stabilize optimization, we also apply decoder-atom $\ell_2$ projection after each step to prevent scale drift and to make purity comparable across runs. Finally, we perform early stopping on validation MSE with a small patience (e.g., 5–10 epochs) and report metrics on both the regular test set and the extreme subset. Unless otherwise stated, Active-# and MSE are computed with unconstrained codes, while Top-$k$ hits are evaluated with $k \in \{1, 5, 10\}$.

## D.3  THRESHOLD SENSITIVITY OF TOP-$k$ HITS

Figure 12 shows how Top-5 and Top-10 hit rates increase with the extremality threshold $p$. Lower $p$ includes more regular samples, while higher $p$ retains cases dominated by a single channel. Both

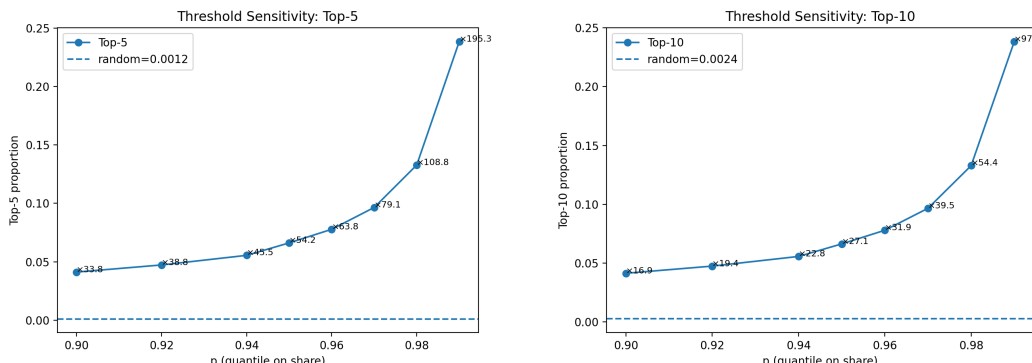

Figure 12: **Threshold sensitivity analysis of SAE Top-$k$ hits.**

metrics rise sharply as $p$ grows, surpassing the random baseline beyond $p \approx 0.98$ and showing near-exponential gains around $p = 0.99$.

This analysis shows that in extreme samples, massive activations align closely with SAE dictionary atoms, indicating that the SAE can automatically extract sparse directions synchronized with max-channel dominance. These findings are consistent with our earlier results: in non-extreme cases, the effect of the max channel is diluted by many competing directions, leading to low hit rates and high redundancy; in extreme cases, its dominance is clearly captured, yielding high hit rates and strong single-axis structure. This provides independent quantitative evidence that massive activations are not incidental anomalies but structural features tightly coupled to the model's readout mechanism. Combined with the log-SNR phase analysis, a coherent picture emerges: as bulk energy rises in later stages, the number of active atoms remains low, while higher Purity-L1 shows that key atoms continue to align strongly with the readout. In contrast, global SAE on regular data exhibits low hit rates, high Active-#, and large reconstruction error, further highlighting that only in the extreme regime do sparse atoms reliably recover the dominant max-channel direction. Altogether, this evidence chain reinforces our central claim: massive activations are not just directional, but also prescriptive signals that guide the model's outputs.

