# OpenReview forum: "Early Guidance, Late Convergence: Hidden‑State Massive Values in Diffusion MLLMs"
_ICLR.cc/2026/Conference — Submitted to ICLR 2026_

### Official Review · Reviewer_ekbM · 2025-10-31

**Soundness:** 3
**Presentation:** 3
**Contribution:** 3
**Rating:** 6
**Confidence:** 4

**Summary:**

This paper examines the effect of large activation values in dMLLMs, attempting to establish a connection between large values and semantic outcomes during generation. First, the authors show a correlation between the channels with the highest value and the channels that affect the logit token the most, showing that these massive values determine tangible generation-time decisions. In addition, massive value channels can spike for certain tokens and not others. The authors also identify movement of massive values across diffusion time, observing a two-phase behavior. They conclude with an SAE analysis of massive channels.

**Strengths:**

- Very nice set of careful empirical investigations that identify unique phenomenon occurring with massive values in dMMLMs.

- Well designed tracking of key metrics across diffusion steps/layers that derive insight as to the evolution of massive values through time and depth.

**Weaknesses:**

- Some of the claims appear to be quite strong/general without full proper empirical backing. For example, the statement about deep layers reinforcing semantic confidence: is there an ablation experiment where reducing the magnitude of massive channels results in generations with higher perplexities? Intuitively massive channels associated with massive logit values would suggest this, but such an experiment would be nice to have.

- Likewise, for the diffusion step-wise masking, the experiments support the claim that massive values at early diffusion steps prevent the model from devolving into incoherence. However, the claims on semantic convergence for massive values in later steps are unclear. Additional discussion that clarifies the types of fine-grained semantics decided by late stage massive values would be much appreciated.

- The claim of massive values being "reusable semantic axes" that are "building blocks" seems overstated with the given SAE analysis and lack of qualitative associations between a massive value and particular semantic effects beyond coherence breakdown.

**Questions:**

- A little more exposition for the final version would be super useful: e.g., explaining what the LogitLens is/why it is a meaningful quantity to look at when you introduce your Agree metric, explaining what the base, front, back, all ablations exactly are, etc.

- In Figure 5, why are the update sizes for a full ablation of similar magnitude to only ablating at late stages? This seems contrary to the claim that update sizes for late stage ablations indicate finer-grained semantic convergence.

- What is the exact claim about relating residual energy update sizes to different scales of semantic convergence (i.e. general coherence vs. fine-grained)?

- The SAE experiments are a little unclear. If you extract an extreme subset, shouldn't you expect the SAE to pick up on it? Analogously, if you run PCA where you curate a dataset consisting of extreme channels, wouldn't the principal components be forced to pick up on these channel directions since the maximum variance is in that direction?

- Typically SAE analyses are accompanied by some kind of semantic association. Are there experiments you can run with the SAE that associate massive channel directions to particular semantic effects in generation?

---

### Official Review · Reviewer_im72 · 2025-11-01

**Soundness:** 1
**Presentation:** 3
**Contribution:** 2
**Rating:** 2
**Confidence:** 3

**Summary:**

This paper investigate the behaviors of diffusion multimodal LLMs (dMLLMs) on hidden-states massive values (extremly large values). Specifically the authors study the massive values in different diffusion steps and transformer layers, and conclude several findings including: 1) massive values correlate to token confidence; 2) massive values across layers and steps can affect the model's outputs; 3) massive values are semantically meaningful in the representation space.

**Strengths:**

1. This paper studies an important prblem that aims to understand the internal behaviors of dMLLMs.
2. The motivation is well introduced so that the audience can easily follow the problem this paper is focusing on.
3. The authors raises several important and meaningful questions that are useful for understanding dMLLMs.

**Weaknesses:**

Although the raised questions are interesting, this manuscirpt is not ready for publication with the following limitations:
1. Main concern: the detailed experimental setting such as detailed experimental methods, the datasets used for this invesitigation, and the data size used for demonstrating results are missing. Hence, serveral insights in this paper are not convincing enough. This concern is related to some of the specific points below.
2. The conclusion in Section 3.1 is not clear, what's the meaning of the color in figure2, how many examples are used and how the final values are aggreated? And it seems only a single example is selected for demonstarting the results in Figure 3c. It could be problematic if we conclude the results from a single example without clear justrification.
3. In line 265: it is problematic to conclude that typucal channels are flat as massive values and smaller values are shown in the same scale in Figure 3c. The relative changes could be omiited as those channels are selected with significant scale differences. Instead reporting relative changes for massive values and typical channels seperately can be solid.
4. There are still gaps between the obsearvations in section 3 and 4, and the conlusions made without more theoritical or emprical support. For example, the different behaviors on steps and layers are not directly related to the semantics in representation spaces.



Other comments:
1. Line 87 and Line 133: missing space between two sentences.
2. Line 117: missing citation.
3. Massive value is formally defined in line 205 of section 3.2, while it is nessassary for explaning the experimental methods from section 3.1. Otherwise it could be hard to fully understand results in figure 2.
4. Figure 3c is almost not readable. Higher quality of the image would be helpful.

**Questions:**

1. What's the actual meaning of $c_j(s,l,t)$ in line 212? I would appreciate the clarification of the reason calculating the contribution in this way.
2. In section 4 the massive values are manipulated to confirm that the imporatance of them. However, it's a bit straightforward to me that manipulating values in hidden states can affect the final outputs. Why don't we test manipulating non-massive values too for ablation?

---

### Official Review · Reviewer_uYRh · 2025-11-03

**Soundness:** 2
**Presentation:** 3
**Contribution:** 2
**Rating:** 4
**Confidence:** 4

**Summary:**

This paper investigates the massive value that exists in the dMLLMs, where some activations are extraordinarily large and consistently appear across layers and timesteps. The authors further analyze that those values align closely with the semantics and confidence. Massive values would serve as a signal in the early stage as the global information, and as a semantic signal for output logits in the later stage.

**Strengths:**

1. The paper investigates the problem of massive value in dMLLMs, which may inspire further research for quantization or optimization of dMLLMs.
2. The paper has several interesting insights, such as in the early stage and the later stage, the massive value would have very different effect on the output.
3. The conclusion that massive value is related to the semantics convergence and also the output confidence is interesting.

**Weaknesses:**

1. The paper only investigates this phenomenon, but does not discuss or show any experiments on how this can help the dMLLMs. It would be better if the authors could have a section to discuss how this would affect the performance/efficiency or anything that is important for dMLLMs.
2. The experiments are only conducted upon one dMLLM (LaVida). It's not clear if this phenomenon is a common one or only a special property of this model.
3. The massive value problem actually has no relationship with the multi-modality part of this model. Did the authors also try to analyze this  on dLLMs, such as LLaDA/Dream?
4. Some experiments cannot fully support the conclusion the author wants to claim. For example, the author wants to claim that the massive value guides the global generation skeleton in early noisy steps. However, no experiment can directly show that it actually impacts the global representations. The error in the early stage would propagate to all the steps later, and thus a small error in the early stage would have larger impact than the one on the later stage, and I think it cannot directly show that it's related to the global generation skeleton.

**Questions:**

See Weaknesses

---

### Meta-Review · Area_Chair_7nyk · 2025-12-18

**Summary:**

The paper investigates the phenomenon of "massive values", extraordinarily large hidden state activations, in diffusion Multimodal Large Language Models, analyzing their evolution across layers and timesteps . While the reviewers acknowledged the motivation was clear and the problem formulation interesting , the consensus is that the manuscript is not ready to be published at this time. The primary concerns preventing acceptance include the reliance on a single model without testing broader applicability to other architectures, a lack of experiments demonstrating the practical utility of these findings for model optimization, and missing experimental details regarding datasets and aggregation methods. Furthermore, reviewers felt that causal claims regarding semantic convergence and global skeletons were overstated and lacked sufficient empirical evidence. Based on these unresolved issues regarding soundness and validation, the recommendation is to reject.

**Reviewer Concerns:**

The authors didn't submit a rebuttal so all raised concerns still remain.

  - The analysis is restricted to a single architecture, leaving it unclear if this is a general MLLM phenomenon or model-specific.
  - There is no demonstration of how these insights improve performance, efficiency, or quantization.
  - The claims that massive values "guide global structure" in early steps or represent "reusable semantic axes" are viewed as correlational rather than causal, lacking sufficient ablation support.
  - Missing details on datasets, data size, and aggregation methods, alongside unclear visualizations hinder reproducibility and soundness assessments.

**Reviewer Scores:**

Same scores since no rebuttal was submitted.

---

### Decision · Program_Chairs · 2026-01-26

Reject